# Neighborhood-Level Lead Paint Hazard for Children under 6: A Tool for Proactive and Equitable Intervention

**DOI:** 10.3390/ijerph18052471

**Published:** 2021-03-03

**Authors:** Mikyung Baek, Michael B. Outrich, Kierra S. Barnett, Jason Reece

**Affiliations:** 1Kirwan Institute for the Study of Race and Ethnicity, The Ohio State University, Columbus, OH 43201, USA; outrich.1@osu.edu (M.B.O.); barnett.433@osu.edu (K.S.B.); 2City & Regional Planning, Knowlton School of Architecture, The Ohio State University, Columbus, OH 43210, USA; reece.35@osu.edu

**Keywords:** lead poisoning, lead hazard risk, lead-based paint, place-based approach, social determinants of health, racial health disparity, child-friendly planning, public health

## Abstract

Lead is well known for its adverse health effects on children, particularly when exposure occurs at earlier ages. The primary source of lead hazards among young children is paint used in buildings built before 1978. Despite being 100% preventable, some children remain exposed and state and local policies often remain reactive. This study presents a methodology for planners and public health practitioners to proactively address lead risks among young children. Using geospatial analyses, this study examines neighborhood level measurement of lead paint hazard in homes and childcare facilities and the concentration of children aged 0–5. Results highlight areas of potential lead paint hazard hotspots within a county in the Midwestern state studied, which coincides with higher concentration of non-white children. This places lead paint hazard in the context of social determinants of health, where existing disparity in distribution of social and economic resources reinforces health inequity. In addition to being proactive, lead poisoning intervention efforts need to be multi-dimensional and coordinated among multiple parties involved. Identifying children in higher lead paint hazard areas, screening and treating them, and repairing their homes and childcare facilities will require close collaboration of healthcare professionals, local housing and planning authorities, and community members.

## 1. Introduction

Lead exposure is a serious health risk for children because of its adverse effects such as damage to the brain and nervous system, slowed growth and development, or learning and behavior problems [1,2,3,4,5,6]. These adverse effects of lead exposure at early ages can bring long-term consequences later in life such as encounters with the juvenile justice system, antisocial behaviors, or impacts on socioeconomic status [7,8,9,10]. The US has a long history of policies and efforts addressing lead poisoning dating back to 1971’s Lead-Based Paint Poisoning Prevention Act and 1978’s residential lead-based paint ban as the country saw a sharp increase in children’s blood lead levels caused by the widespread lead contamination in the early 1970s [11]. In 1992, Congress passed the Residential Lead-Based Paint Hazard Reduction Act of 1992, also known as Title X of the Housing and Community Development Act. This law directed US Department of Housing and Urban Development and US Environmental Protection Agency to require the disclose of known information on lead-based paint and lead-based paint hazards before the sale or lease of most housing built before 1978 [12]. By 2009, the Surgeon General of the US Public Health Service issued *“The Surgeon General’s Call to Action to Promote Healthy Homes”* to provide an overview of issues contributing to the nation’s unhealthy housing situation and to draw attention to the public health impact of housing hazards [13]. Decades of these efforts are considered a major public success as it resulted in a 93.6% decline in geometric mean blood lead level of the US population from 12.8 to 0.92 µg/dL between 1976 and 1980 to 2015 and 2016 [12]. Contrary to a contemporary misconception that lead exposure is no longer a problem, the Centers for Disease Control and Prevention estimates that approximately 530,000 US children aged 1–5 years or 2.6% of children in this age group had ≥5 µg/dL from the periods 1999–2002 to 2007–2010 [14]. Despite these improvements, previous research stresses that there is no safe blood lead level in children and even very low level of lead exposure can cause health impairment [1,2,3,4,10].

Children get exposed to lead from multiple sources including air, bare soil, food, drinking water, and/or consumer goods. However, the primary source of lead exposure in children is deteriorating lead-based paint found in older homes and buildings as well as contaminated dust and soil, which accounts for up to 70% of elevated blood lead levels in children in the US [15]. The Centers for Disease Control and Prevention states that lead poisoning is 100% preventable and identifies four strategies to address this public health concern: (1) getting a blood lead test of your child, (2) testing paint and dust in your home for lead, (3) renovating your home, and (4) removing lead-containing products from your households [16]. However, lead-based paint hazards remain to be a critical environmental hazard in many homes of children across the United States. The results of the American Healthy Homes Survey, conducted between June 2005 and March 2006, estimated that over 23 million homes or 22% of the nation’s housing still had significant lead-based paint hazards. It was also reported that 3.6 million of the homes belonged to households with one or more children younger than six years of age [17]. This number could have been lowered with recent lead poisoning prevention efforts, but as of 2010, homes of 530,000 children with elevated blood levels could potentially have lead-based paint hazards in them [14]. Additionally, the risk of lead exposure is not evenly distributed across the US population, with greater risks for communities of color and residents in urban and low-income areas where older, lead-contaminated homes are located. Studies reported consistent and significant racial disparity in lead exposure across the nation with higher blood lead levels for non-Hispanic Black children compared to non-Hispanic whites even after correcting for other housing or socioeconomic risk factors [14,18,19,20,21]. Therefore, additional strategies need to be deployed to remedy these racial disparities in lead exposure.

Federal, state, and local lead policies to date have been mostly passive and reactive, concentrating its efforts on mitigation or remediation of lead exposure after-the-fact using children’s elevated blood lead level as an indicator to act upon. In addition, persistence of inequities in lead exposure to this date reveals the limitations of lead policies in lacking their focus on health equity [14,18,22]. Future approaches to lead exposure interventions need to be proactive and prevention-oriented and should be multidisciplinary and well-coordinated among multiple parties involved. Most importantly, these new approaches must focus on eliminating health inequities related to lead [18,22].

The first critical step in establishing effective approaches to lead exposure prevention for children is to accurately identify target intervention areas. This allows public health practitioners to better pinpoint their efforts by locating neighborhoods where higher level of lead hazard is suspected and also more children of younger ages spend most of their time. In this study, we present a methodology for the measurement of child-focused lead-based paint hazard using geospatial analyses that will facilitate effective targeting of lead hazard areas at neighborhood level. This entails examining concentration of children aged 0–5 and conditions of two major built environments where younger children live, learn, and play—homes and childcare settings [19].

Incorporating community-level data for effective identification and mitigation of health risks around one’s neighborhood is important [23]. One’s health is largely determined by social determinants of health, defined as “the circumstances in which people live, work and grow; largely shaped by the distribution of resources and power” [24]. The distribution of such resources is uneven across different neighborhoods, resulting in the concentration of disadvantage in certain areas and health disparity that is very much place-based. With the use of community-based data on neighborhood housing, childcare centers, and distribution of children by race and ethnicity, this study exemplifies the merits of a place-based approach and the use of Geographic Information System in healthcare and health intervention [23,25,26]. The lead paint hazard calculated at census block group in this study will help better facilitate lead poisoning prevention efforts by identifying areas where more buildings require renovation, and more children require blood lead tests. This will also promote cross-sector collaboration among multiple parties from public health, local housing, and pediatric clinicians. More importantly, analyses of distribution of children of different race and ethnicity in relation to lead paint hazard will address racial health disparity in lead exposure and specifically identify areas to target interventions to address those disparities.

## 2. Materials and Methods

### 2.1. Study Area

This analysis explored the geographic distribution of lead hazard risk in residential parcels across Franklin County, Ohio (Figure 1). As Ohio’s most populous county, with over 1.3 million people, 54.5% of all housing units in Franklin County were built before 1980. This means that over half of housing units in the county are susceptible to having lead paint in the home, as lead paint was outlawed in 1978 [27].

### 2.2. Data

#### 2.2.1. Franklin County Auditor Data

This study used parcel level data from the Franklin County Auditor to assess the age, value, and quality of residential parcels and associated improvements across Franklin County. Because this analysis quantifies the level of lead hazard risk, residential parcels built after 1978 were excluded since the hazard would not be present in them. Of the 383,716 residential parcels in Franklin County, 181,986 (47.4%) residential parcels were built after 1978 and therefore excluded from the analysis.

#### 2.2.2. US Census Tiger Line Shapefiles

This analysis included the use of three geographic units—census block, census block group, and census tract—acquired from the US Census Bureau. Census block, the smallest of the three, provides the finest geographic resolution and detail, while census block group allows for a basic overlay analysis to assess the level of exposure to lead paint risk at home among children under 5 in Franklin County. Census tracts provide only an average level of detail at the neighborhood scale to assess lead risk. However, census tracts allow for the overlay analysis to assess the level of Franklin County children under five exposure to lead paint risk at home and examine the exposure risk by race.

#### 2.2.3. American Community Survey 5-Year Estimates (2015–2019) Data

At the Census tract level, from ACS 5 Year estimate tables B01001A–I: Sex by Age (by Race), the totals for each racial group Under 5 Years of age by sex were summated to achieve the total number of children Under 5 for each race for all Franklin County census tracts. At the Block group level, from ACS 5 Year Estimate table A01001: Age, the Under 5 column was selected for use in this analysis because there was no data available disaggregated by race at the block group level.

#### 2.2.4. Ohio Certified Child Care Center Data

Certified childcare centers in Franklin County, Ohio, were downloaded from the Ohio Department of Job and Family Services-Child Care in Ohio website, which resulted in 1079 records. These included childcare centers of different types in the county: licensed childcare centers, licensed family childcare homes, Ohio Department of Education licensed preschool and school age childcare, and registered day camps. After geocoding the addresses of the childcare centers, 3 records were determined to be located outside of this study area and thus excluded from the analysis.

### 2.3. Methods

Figure 2 provides an overview the methodology used to develop the lead risk scores. This study used parcel level data obtained from the Franklin County Auditor to assess the age, value, and quality of residential parcels and associated improvements. After acquiring all residential parcels, all parcels built after 1978 were excluded. Next, parcels were categorized based on appraised property values. Lower valued properties are most likely to be in disinvested communities and thus have a higher risk of deferred maintenance which increases the risk of lead dust from paint. Valuations were classified into three tiers: (1) high (values > $200,000), (2) moderate (values $100,000–$200,000), and (3) low (<$100,000). Parcels classified with a low valuation were assigned a risk valuation of 3, those in the moderate were assigned a risk of 2, and high valued properties were assigned 1. Next, the age of the property was factored into the risk methodology. Properties were classified into three categories based on the likelihood of containing lead-based paint in the home considering the age of the home [28]. Oldest properties built before 1940 were assigned a risk value of 3, moderate age properties built after 1940 but before 1960 were assigned a risk value of 2, and newer properties built after 1960 were assigned a value of 1. A cumulative score was then calculated based on the parcel attributes and risk score ratings.

#### 2.3.1. Evaluation Building Grade Quality Risk

Data from the Franklin County Auditor also included a building grade for all residential properties that refer to the quality of construction. The grades range from E-- (Lowest) to A++ (Highest). Lower graded properties are more likely to be built with material that are substandard and are likely to decay quicker. Grades D and E properties are homes that have visible disrepair and while habitable, are the most likely to be environments where lead dust would be in the air. Lead dust is inferred based on deferred maintenance prevalent in D and E grade properties, hence the higher risk for lead dust and poisoning and higher modifier [29]. Grade C refers to properties built with materials and construction quality that are average in stature and have normal wear and tear. Grade B properties are homes built with stronger materials and finishes or have been renovated and updated recently with above standard materials. Grade A properties are homes that are usually custom built and have the highest end materials and finishes. The criteria used to assign grades are shown in Table A1 in Appendix A. A risk modifier, based on grade, is multiplied by the cumulative score to obtain the lead risk score for each residential property. The modifier comes from the “Replacement cost” calculations but normalized to 0 based on the average C grade [29]. Modifiers and total counts of parcels in the county can be found in Table A2 in Appendix A.

Given the age and value of the property allows us to identify the locations where disinvestment is most likely to occur, the scoring alone does not tell us information about the quality of those buildings or the likelihood they would be in disrepair. As a result, applying the grading modifier can help us control for this limitation. Since studies show lead particles mostly come from peeling lead paint, the lower the property grading the more likely peeling paint would be noticed by the appraisers; and if the property is valued lower and older, it is more likely to not have been reinvested in recently or perhaps ever. Table 1 depicts an example of calculations used to develop the cumulative score for each example property and the final score after the risk modifier is applied.

Once the individual property lead risk scores are calculated, a cumulative lead risk score can be aggregated based on any geographic scale. For this analysis we first aggregated the residential lead risk scores to census blocks. To ensure the lead risk score is adequately represented in the context of all residential development in a neighborhood, all residential parcels within each block are summated and then divided by the total number of residential properties in the block. This includes residential properties built after 1978 which have a lead paint risk score of 0. The same process is conducted for the census block group and census tract geographic units. The area lead risk score is then classified into five categories based on Table 2.

#### 2.3.2. Assessing Community Risk at Childcare Centers

To assign a neighborhood lead risk to childcare centers, all childcare centers in the county are geocoded and mapped with the residential parcel lead risk data. A ½ mile buffer around each childcare center was used to approximate access. This assumes all residents who need childcare utilize the nearest childcare center. All residential parcels and associated lead risk scores are summated within the ½ mile of each childcare center and then normalized by all residential parcels in the area. Each center’s score is then classified into five categories using the same score ranges used to obtain the neighborhood risk score. Childcare centers within each risk category are classified in Table 3.

## 3. Results and Discussion

### 3.1. Lead Risk in Franklin County and Child Concentration

The results of lead risk score calculated for individual properties and aggregated by census block and block group are mapped for Franklin County, Ohio, in Figure 3 and Figure 4, respectively. These maps show the distribution of lead paint hazard for neighborhoods across the county. Census block (and block group) areas where there are higher proportions of properties of higher lead risks for residential properties within the neighborhood (i.e., neighborhoods with higher lead risk scores) are depicted in red and darker red in the map. It is notable that areas of higher lead hazard (Very High and High lead risk scores) are concentrated in the central part of the county whereas lower lead risk neighborhoods (Very Low and Low lead risk scores) are dispersed in outer parts of the county, particularly for Very Low lead risk areas, shown in light blue. Given Columbus’ aggressive annexation policies that started in the 1950s, most of the lead risk hazard exists within the City of Columbus. The only exception would be in the suburb of Whitehall which was built during the 1950s and 1960s but has experienced disinvestment and decline since the 1990s.

By taking multiple characteristics of residential parcels—built year, property value, and building quality—into account, this lead risk score captures a more holistic measure of lead-based paint hazard in residential settings. Visualization of neighborhood lead risk on a map helps identify areas of differential lead risk across the county and highlight potential areas where immediate intervention is required. These are areas where county and city planners can strategically target communities for home renovation and remove lead-based paint to remove lead poisoning hazard from built environment in the area. Additionally, it is important to note that the level of aggregation that planners choose to visualize the data will ultimately impact the precision of their targeted interventions. For example, when comparing the maps shown in Figure 3 and Figure 4, many of the high and very high risk areas that lay on the outskirts of the county in Figure 3 are lost in Figure 4 and classified as low risk. Therefore, visualization at the Block level would provide the finest level of risk to be used to determine targeted intervention areas.

The areas with higher lead risk become a greater health challenge for children if more children reside in the area. Figure 5 is a map of neighborhoods with higher lead risk (Very High and High) where relatively more children live compared to other areas in the county (High and Moderate consideration of children).

By adding a layer of data depicting the population of children to the lead risk distribution, this map can be useful tool for child-focused public health interventions and policymaking. For pediatricians planning lead poisoning prevention efforts, this map provides target areas to administer blood lead tests among children in areas with higher lead risk and higher concentration of children. This aligns with previous research that showed benefits of a place-based approach and the use of community data as ‘geomarkers’ or ‘community vital signs’ for community health risks into population health improvement [23,30,31,32,33]. While existing studies used community data at census tract or census block group level, this study improves the correctness of neighborhood lead risk assessment by using parcel-level data to take into account lead risk for individual property in measuring risks for lead-based paint in residential buildings.

Table 4 shows the total number of children in neighborhoods with different lead risk levels based on our analysis. Alarmingly, close to 25,000 children under 5, or 27.5% of children in that age group, reside in areas of either high or very high lead risk level. The numbers for children under 18 is over 80,000 or 27.1%. The fact that more than one out of four children in the county live under potential lead poisoning risk is an urgent call for public health and local policymakers to make more aggressive efforts, such as more testing and more housing renovation, to mitigate this risk.

Figure 3, Figure 4 and Figure 5, used in combination, have a potential for assisting with making more proactive and comprehensive child-focused lead poisoning prevention policies by guiding streamlined efforts such as blood lead testing or home remediation. While city planners could use lead risk maps (Figure 3 and Figure 4) to focus housing renovation program to census block areas of higher lead risk, health professionals, in collaboration with community organizers, could benefit from Figure 5 in implementing more aggressive blood lead testing in areas with higher lead risk and more concentration of children. In the process, the place-based approach suggested here can also support cross-sectional collaboration in the identification of multiple parties to be involved—parents, caregivers, healthcare providers and professionals, community organizers, and local housing authorities—and development of partnerships among them, with a shared goal of improving the health of children in the neighborhoods [23,34,35,36,37].

### 3.2. Racial Disparity in Lead Risk for Children

Figure 6 shows concentration of non-white children in relation to differing level of lead risk at census tract level in Franklin County, Ohio. It should be noted that lead risk scores were aggregated at census tract in these two maps to enable cross-comparison with number of children by racial group, an estimate only available at census tract level. Furthermore, 79.4% of very high or high lead risk census tracts are more than 50% non-white children under 5, with only 4% of tracts with 10% or less non-white. The concentration of non-white children is the complete opposite in the map of low lead risk; 24.5% of low lead risk tracts are more than 50% non-white children under 5, with 6% of tracts having greater than 75% non-white children.

Racial disparity in lead risk in the county is also depicted in Figure 7 below. Among children under 5, close to two thirds of white children and over 75% of Asian children live in areas of low lead risk, as opposed to 45% of Black children and 37% of Hispanic children that live in areas of high or very high lead risk. The number of Black children in high lead risk areas (*n* = 11,484) takes up about half of total number of children under 5 living under higher lead risk levels (*n* = 24,486) in the county, meaning one of two children aged 0–5 living under high lead risk areas in the county are Black, while they take up only 27.8% of total children under 5.

### 3.3. Lead Hazard, Social Determinants of Health, and Opportunity Indices

The uneven distribution of lead risk for children of different race in Franklin County is not unique and is just one example of inequitable environmental health hazards that leads to health disparity across racial lines, disproportionally harming people in minority communities. Growing up in neighborhoods with greater health disadvantages, Black and Brown children become more vulnerable to various health risks and eventually more prone to illness. The residential segregation found across the country plays an important role in reinforcing such disadvantages for communities of color. Such communities become more susceptible to environmental hazards with disproportionate siting of environmental hazard facilities close to their neighborhoods as pollution is known to be correlated with race and poverty lines [38,39,40]. Higher level of lead risk in neighborhoods with more older housing units often coincides with concentration of non-whites as well as other socioeconomic factors such as income level, parental occupations, or housing tenure types [11,15,21,41]. A prolonged exposure to the combination of these factors, also referred as ‘social determinants of health,’ impacts medical conditions of children and determines their long-term health conditions. Noting the collective impact of such factors as root causes of various medical problems, health professionals call for screening of these factors that goes beyond examination rooms as wells as a systemic evaluation of local resources to improve health and reduce disparities [42,43].

Because neighborhood lead risk is correlated with socioeconomic factors and social determinants of health, addressing lead risk might result in addressing other health risks in the neighborhoods. To examine how lead risk is related to other factors associated with health of children, we also compared lead risk scores with two other measurements of neighborhood resources, Child Opportunity Index 2.0 [44] and Health Opportunity Index [45]. Child Opportunity Index 2.0 (COI) is a composite metric of 29 neighborhood conditions affecting healthy development of children across three domains: education, health and environment, and social and economic. Health Opportunity Index (HOI), a composite measure derived based on 13 variables, is a tool designed for understanding the effect of social determinants of health on health outcomes.

Because COI and HOI are calculated by census tract, lead risk score for Franklin County was also calculated at the census tract level for proper comparison. Figure 8 and Figure 9 compares maps of lead risk, COI, and HOI and Figure 10 shows the association between lead risk scores, COI and HOI, respectively. Note that higher scores of lead risk denote higher risk for lead, while higher scores of COI and HOI denote higher child opportunity and higher health opportunity, which are positive. Both COI and HOI are negatively correlated with lead risk with R-squared values of 0.62 and 0.53. While HOI has lower R-squared value, the scatter plot shows better fit compared to COI, which might be contributable to the multiple domains COI covered compared to health-specific variables used for HOI.

In addition to assessing the correlation between lead risk scores and COI and HOI scores, the distribution of children of different race in relation to differing levels of lead risk, COI and HOI are examined, shown in Figure 11. The racial disparity in lead risk distribution in the county, noted with disproportionally higher number of Black and Hispanic children in higher lead risk neighborhoods, is also noted in relation to COI and HOI, where higher proportion of Black and Hispanic children have a lower level of COI and HOI (Very Low and Low).

The analysis of similarity and co-occurrence between lead risk, COI and HOI showed that areas with higher lead risk are also areas with other health risks (lower HOI) and lower child opportunity areas (lower COI). It also suggests that addressing neighborhood lead risk by administering more blood lead tests for children and fixing houses in high lead risk areas will also result in improving level of opportunity for children in those communities and overall health opportunity for people of all ages. The result of racial analysis also suggests lead poisoning prevention efforts, focused on high lead risk areas, will eventually facilitate addressing racial disparity not only in lead risk but also advance overall health inequity and child opportunity across the county.

### 3.4. Lead Risk and Childcare Centers

In addition to where children live, another important surrounding where children spend a lot of time is childcare facilities, considered as an extension of home for children of younger ages. Therefore, any lead poisoning prevention plan needs to ensure including children care facilities in their efforts. With close to 140,000 children currently attending various types of childcare and lack of statewide requirement for lead testing of schools or childcare facilities, ensuring lead safety in childcare is a critical part in lead poisoning prevention policies [19]. To identify childcare facilities located in neighborhoods with higher lead risk in Franklin County, locations of all childcare facilities in the county are overlaid on the lead risk map (Figure 12) and counts of childcare programs in different lead risk areas are summarized in Table 5.

It is essential that efforts address the lead risk both in the home and at childcare facilities, a strategy that the American Academy of Pediatrics also recommends [46]. Consider children who live in neighborhoods with lower lead risk, but attend childcare centers located in higher lead risk neighborhoods. While their risk for lead exposure is controlled in their homes, spending considerable amount of their daytime at their childcare centers may expose them to lead risk, particularly if they are attending family childcare homes. In addressing lead risk for children, childcare centers can act as critical testing sites for detecting potential lead risk outside homes.

A recent study has called for the collaboration between healthcare providers, local governments, and schools to increase the availability of services for children [47]. A collaboration between healthcare providers, local governments or public health officials, and childcare centers could result in additional testing resources being brought into these facilitates. This analysis will not only assist with identifying locations of childcare centers in higher lead risk areas, it will also facilitate remedying different types of childcare programs depending on organizations overseeing them. This will allow the local governments to establish policies that will ensure that the cross-agency collaborative efforts can be executed across the varying types of childcare facilities.

### 3.5. Limitations

This study has a few limitations. Among multiple sources of lead exposure for children such as air, soil, food, or drinking water, this study analyzed lead risk from lead-based paint only. By using data for residential buildings in estimating lead risk, this study fails to account for lead exposure from other environmental sources such as nearby polluting industries. In addition, this study does not evaluate lead paint risk in large scale apartment communities, which are not classified as ‘residential’ per Franklin County Auditor data. The study does include condominium properties or scattered site smaller scale apartment communities but does not evaluate the lead risk that could vary from unit to unit within an apartment community. Because of this limitation, there are three census tracts and four census block groups in the analysis that display “No data”.

Another limitation of this study is that due to lack of data availability, the methodology could not be validated using data for lead test in buildings or blood lead level test of children in the County. However, a previous study conducted in Toledo, Ohio confirmed the link between age and value of the residential properties and lead test results for children aged 0–72 months between 2010 and 2014. The area with a higher concentration of older and lower-valued properties was found to have a significantly higher positive elevated blood lead level test rate [48]. Therefore, the methodology presented in this study is expected to approximate communities of interest where testing efforts should be better targeted. Despite the limitations, this is the first study to the authors’ knowledge that utilizes parcel or housing level data to estimate lead hazard risk.

## 4. Conclusions

The quality of environment where children grow up makes critical and long-lasting imprints on the quality of their lives in various aspects including their physical and emotional health, intellectual development and academic accomplishment, and economic and social wellbeing. Among various sources of health risks found in environments for children, this study focuses on lead paint hazard which poses severe health risks for children. Noting the primary source of lead hazards is lead-based paints used in buildings constructed prior to 1978, this study answers an important call for child-friendly planning to improve public health by influencing the environment where children live and play. Using parcel-level data of all residential units in Franklin County Ohio, this study measures lead risk at neighborhood level by accounting for age, value, and quality of properties.

The result of the measurement and visualization of it shows a potential for assisting more targeted and proactive intervention to prevent lead poisoning in the county by identifying target areas at neighborhood level (census block or census block group). A cross-examination of areas with different lead risk and concentration of children revealed that more than a quarter of children under 5 in the county reside in areas with high or very high lead risk, which is close to 25,000 children. This analysis also suggests cross-sector collaboration among parties in healthcare and housing for multidimensional and comprehensive lead hazard intervention efforts through blood lead testing and home remediation.

Further analysis found racial disparity in lead risk in the county with 45% of Black children under 5 live in higher lead risk neighborhoods, compared to 18% of white children. This suggests an urgent need to address racial disparity in lead risk in order to improve public health in the county. To make our community healthier for our children, we need to ensure that the environment is healthy for all children, regardless of their racial or socioeconomic backgrounds because an environment that is unhealthy for some children is unhealthy for all children and adults in the entire community.

Lastly, it is also important to note that the methodology for estimating lead risk presented in this study is replicable and expandable to other parts of the country if parcel level data (age, value, and quality of residential parcels) are available. The year in which homes are built and the property value are often made public by county or state housing agencies. Additionally, the building quality grade categories used by Franklin County (see Table A1 in Appendix A), which ultimately was used in this study, are one of the standards commonly used across the country. Provided that this data can be obtained, lead risk can be estimated at the parcel level, which can subsequently be aggregated to neighborhood level such as census block or tract. This methodology therefore has great potential for improving targeted and proactive lead poisoning prevention efforts for children beyond the county level such as metropolitan area, state, or even at the national level.

## Figures and Tables

**Figure 1 ijerph-18-02471-f001:**
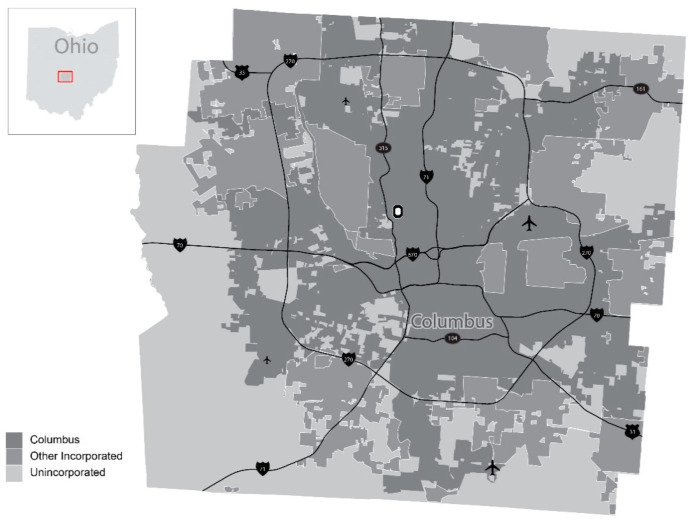
Study area, Franklin County, Ohio.

**Figure 2 ijerph-18-02471-f002:**
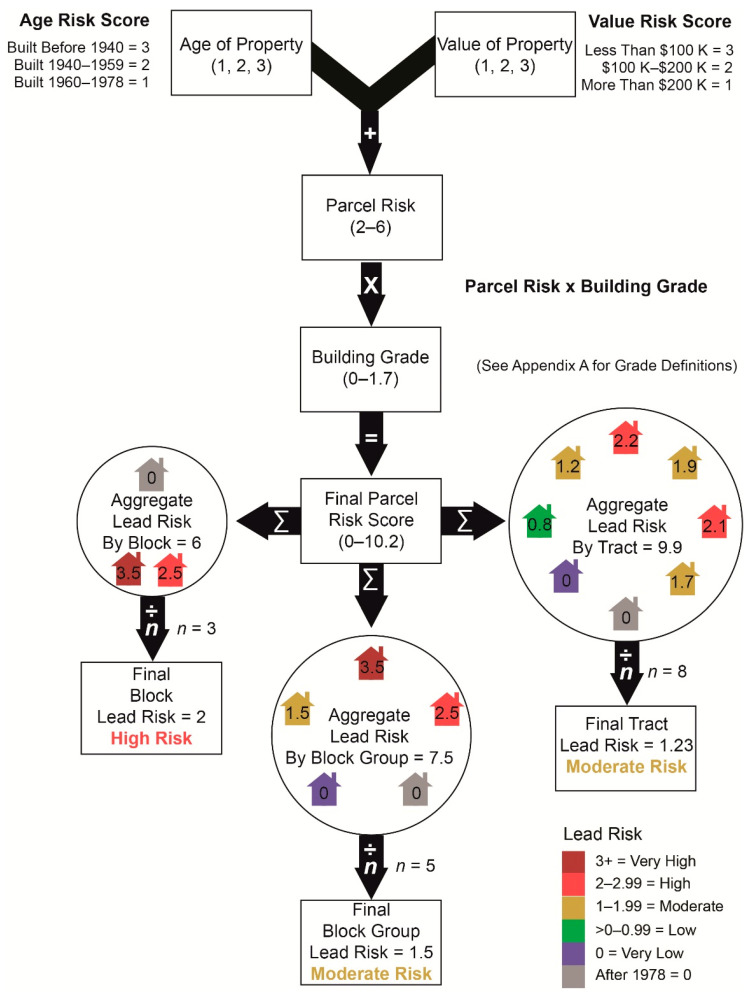
Lead Risk Methodology.

**Figure 3 ijerph-18-02471-f003:**
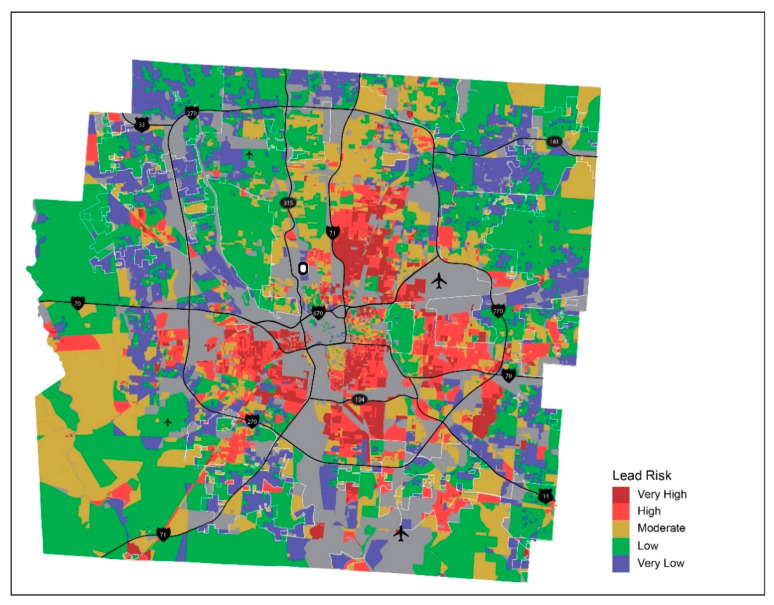
Lead risk by census block—Franklin County, Ohio.

**Figure 4 ijerph-18-02471-f004:**
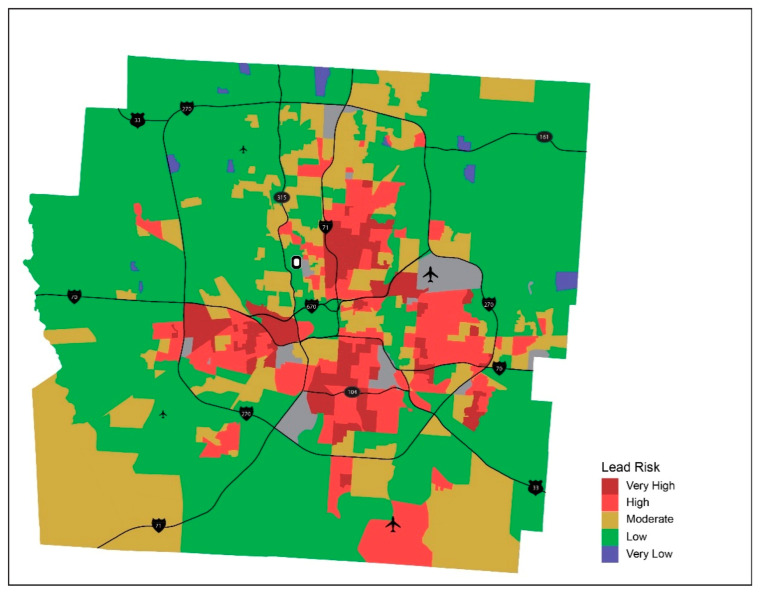
Lead risk by census block group—Franklin County, Ohio.

**Figure 5 ijerph-18-02471-f005:**
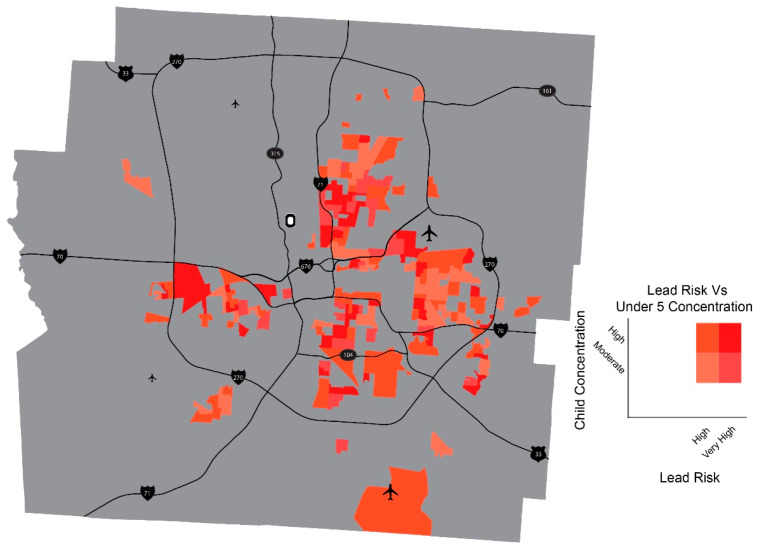
High lead risk areas with more concentration of children under 5.

**Figure 6 ijerph-18-02471-f006:**
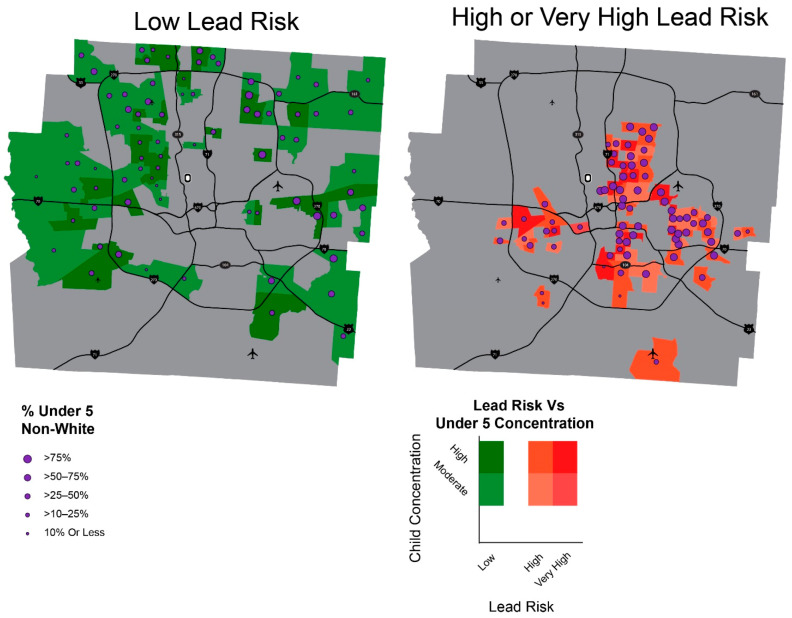
Lead Risk Areas and Percent of Non-White Children.

**Figure 7 ijerph-18-02471-f007:**
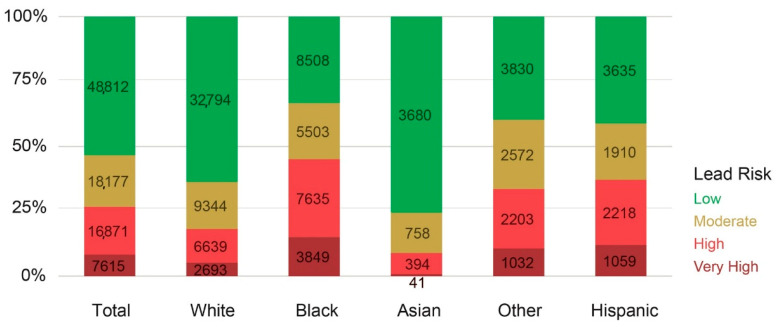
Lead risk and children under 5 by race.

**Figure 8 ijerph-18-02471-f008:**
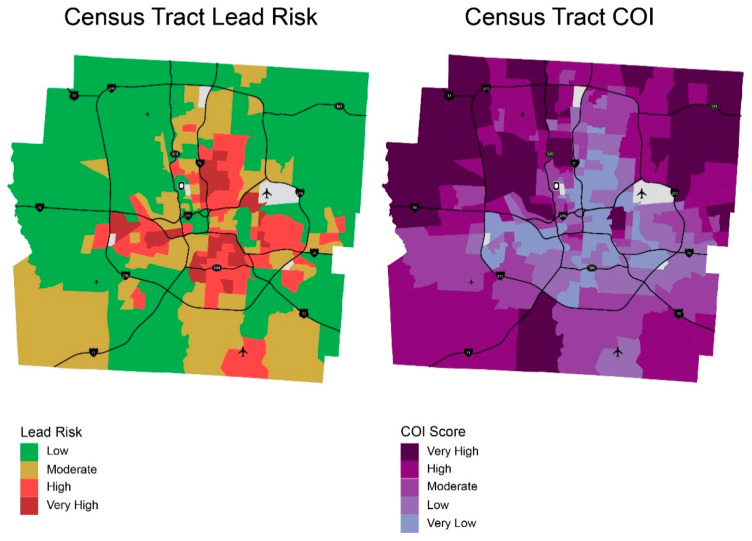
Lead risk versus Child Opportunity Index.

**Figure 9 ijerph-18-02471-f009:**
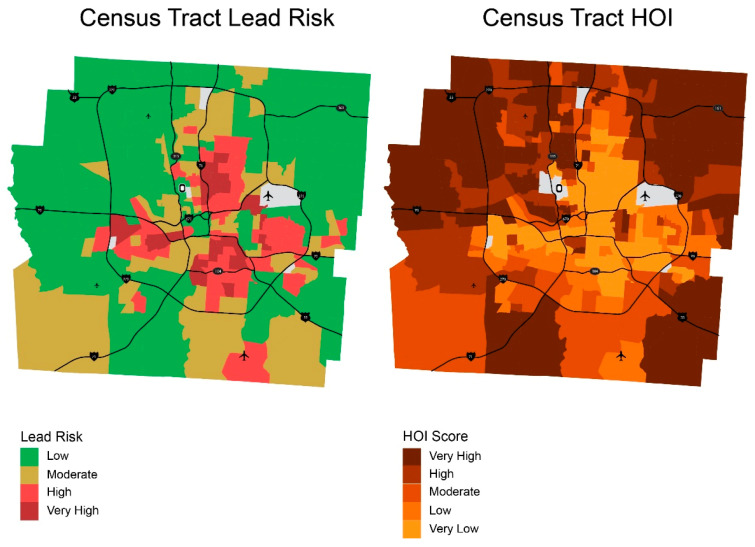
Lead risk versus Health Opportunity Index.

**Figure 10 ijerph-18-02471-f010:**
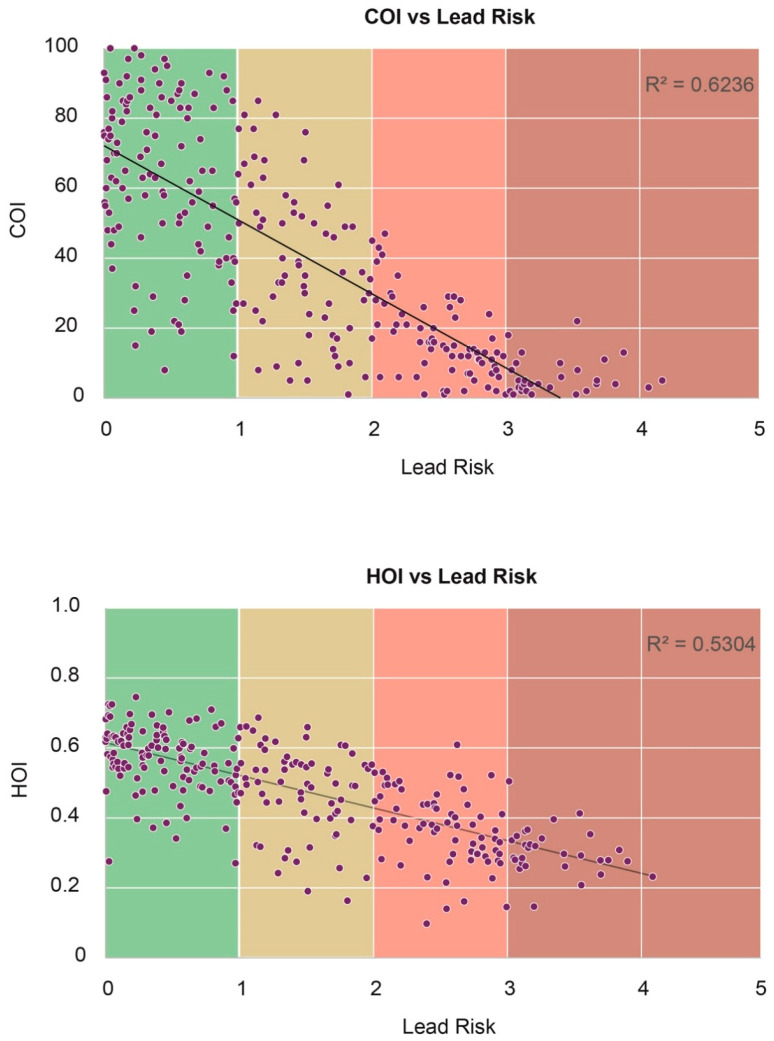
Lead risk compared to Child Opportunity Index 2.0 (COI) and Health Opportunity Index (HOI) score.

**Figure 11 ijerph-18-02471-f011:**
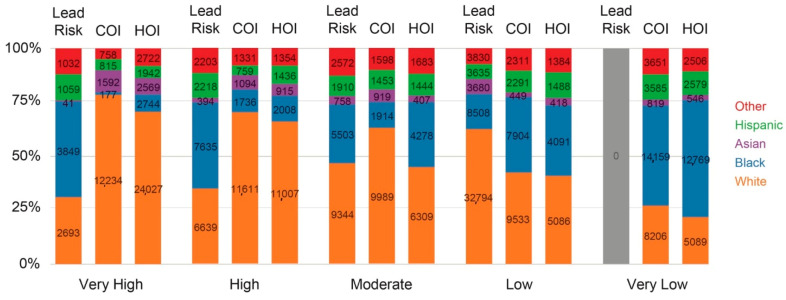
Lead Risk, COI, and HOI by race.

**Figure 12 ijerph-18-02471-f012:**
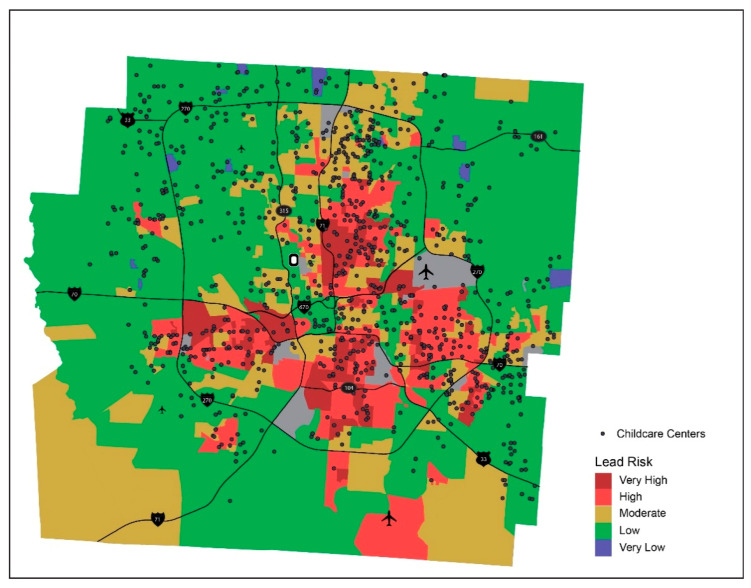
Lead risk and childcare centers.

**Table 1 ijerph-18-02471-t001:** Example of final lead risk score after risk modifier applied.

Examples	Age	Value	Age Score	Value Score	Total Score	Grade	New Score
Property 1	1985	$156,000	Excluded	Excluded	Excluded	Excluded	Excluded
Property 2	1971	$275,000	1	1	2	B	0.2
Property 3	1965	$130,000	1	2	3	C	1.5
Property 4	1961	$70,000	1	3	4	C−	2.4
Property 5	1955	$270,000	2	1	3	B−	0.6
Property 6	1951	$145,000	2	2	4	C−−	2.8
Property 7	1943	$60,000	2	3	5	D+	4.5
Property 8	1932	$265,000	3	1	4	A−	0.1
Property 9	1927	$130,000	3	2	5	C++	1.5
Property 10	1910	$50,000	3	3	6	D	6

**Table 2 ijerph-18-02471-t002:** Quintile classification of lead risk score.

Examples	Score Range	Total Blocks	Total Block Groups	Total Census Tracts
Very High	3.00+	1881	122	30
High	2.00–2.99	2451	190	69
Moderate	1.00–2.99	2897	204	66
Low	0.01–0.99	2803	339	114
Very Low	0.00	3758	13	0
No Data ^1^	NA	0	4	3

^1^ No Data refers to census tracts or census block groups where there are children under 5 living there but without any “residential” parcels identified per Franklin County Auditor data because large apartment communities are not assigned a “residential” classification. Additionally, one of the census tracts and two of the census block groups are on Ohio State University’s campus and in the Franklin County Auditor data, dormitories are considered tax exempt parcels because of Ohio State’s tax exemption.

**Table 3 ijerph-18-02471-t003:** Community lead risk category for childcare centers.

Priority	Risk Score Ranges	Total Centers
Highest	3.000+	96
High	2.000–2.999	241
Moderate	1.000–1.999	283
Low	0–0.999	426
Very Low	0	29

**Table 4 ijerph-18-02471-t004:** Lead risk level and child population in Franklin County, Ohio.

Lead Risk	Total Children under 5	% Children under 5	Total Children under 18	% Children under 18
Very High	9526	10.5%	30,569	10.3%
High	15,361	17.0%	50,084	16.8%
Moderate	17,320	19.1%	52,944	17.8%
Low	45,780	50.6%	157,185	52.8%
Very Low	2494	2.8%	6789	2.3%
Total	90,481	100%	297,571	100%

**Table 5 ijerph-18-02471-t005:** Childcare centers by lead risk level.

Child Care Program Type	Highest	High	Moderate	Low	Grand Total
In Home Aide	0	2	0	0	2
Licensed Childcare Center	52	127	161	296	636
Licensed Type A Family Childcare Home	0	0	2	1	3
Licensed Type B Family Childcare Home	25	63	54	74	216
Ohio Dept. of Education Licensed Preschool	16	34	37	42	129
Ohio Dept. of Education Licensed School Age Childcare	2	13	26	28	69
Registered Day Camp	1	2	3	14	20
Total	96	241	283	455	1075

## Data Availability

Not applicable.

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
