# Peer review of "Neighborhood-Level Lead Paint Hazard for Children under 6: A Tool for Proactive and Equitable Intervention"

_ijerph, 2021, doi:10.3390/ijerph18052471_

Round 1

Reviewer 1 Report

Dear Authors,

This is an excellent, well-written and presented article.  I have made the following suggestions that I think would improve the article.  Thank you for your work.

In line 34, I would recommend replacing the term "juvenile delinquency" with "encounters with the juvenile justice system"

In the introduction, could the authors acknowledge that exposure to lead in water is also relevant, especially for bottle-fed infants.  Additionally, it would be helpful to share anything known about the existence of lead service lines in Franklin County.

Section 2.3.2  Please clarify that this section refers to the multiple types of childcare (center-based, home-based, etc) that are described in section 3.4 and table 5.

In the conclusion, can the authors discuss the availability of data in other geographies in the US and replicability of this method at either the MSA level of the national level?

Author Response

Thank you for the opportunity to submit revisions to our manuscript. We appreciate the time and effort that the editors and reviewers have dedicated to providing meaningful feedback on the manuscript. We have incorporated changes throughout the manuscript to reflect the comments/suggestions provided by the reviewers. Below, please see our responses to your comments and concerns.

Comment: In line 34, I would recommend replacing the term "juvenile delinquency" with "encounters with the juvenile justice system"

Response: Thank you for the suggestion. We have replaced the term accordingly. (See Line 33)

Comment: In the introduction, could the authors acknowledge that exposure to lead in water is also relevant, especially for bottle-fed infants.  Additionally, it would be helpful to share anything known about the existence of lead service lines in Franklin County.

Response: We discussed multiple sources of lead exposure for children in Lines 55-58, including drinking water. Since the focus of the manuscript is investigating lead exposure through lead-based paint found in older homes, we did not include detailed discussion of other sources of lead such as air, soil, or water. Along the same logic, we think that adding the discussion of lead service line in Franklin County might distract readers from the aim/focus of this manuscript.

Comment: Section 2.3.2 Please clarify that this section refers to the multiple types of childcare (center-based, home-based, etc.) that are described in section 3.4 and Table 5.

Response: We have edited Section 2.2.4, which discusses the data, and added the following for clarification:

Lines 148-150: “These included childcare centers of different types in the county – licensed childcare centers, licensed family childcare homes, Ohio Department of Education licensed pre-school and school age childcare, and registered day camps.”

Additionally, we made edits to Section 2.3.2 for clarification as well:

Lines 217-218: “To assign a neighborhood lead risk to childcare centers, all childcare centers in the county are geocoded and mapped with the residential parcel lead risk data.”

Comment: In the conclusion, can the authors discuss the availability of data in other geographies in the US and replicability of this method at either the MSA level of the national level?

Response: Thank you for the suggestion. We have added discussion of replicability and expandability of our methodology in the Conclusion:

Lines 461-471: "Lastly, it is also important to note that the methodology for estimating lead risk presented in this study is replicable and expandable to other parts of the country if parcel level data (age, value, and quality of residential parcels) are available. The year in which homes are built and the property value are often made public by county or state housing agencies. Additionally, the building quality grade categories used by Franklin County (See Table A1 in Appendix A), which ultimately was used in this study, are one of the standards commonly used across the country. Provided that this data can be obtained, lead risk can be estimated at the parcel level, which can subsequently be aggregated to neighborhood level such as census block or tract. This methodology therefore has great potential for improving targeted and proactive lead poisoning prevention efforts for children beyond the county level such as metropolitan area, state, or even at the national level.”

Reviewer 2 Report

A review is attached

Author Response

Thank you for the opportunity to submit revisions to our manuscript. We appreciate the time and effort that the editors and reviewers have dedicated to providing meaningful feedback on the manuscript. We have incorporated changes throughout the manuscript to reflect the comments/suggestions provided by the reviewers.

Below, please see our responses to your comments and concerns.

Comment: The results are well defined but they lack to be separated from the discussion.

Response: The journal allows authors to combine the results and discussion sections, which the authors felt to be more appropriate way of discussing the results of this manuscript. (See instruction for manuscripts here: https://www.mdpi.com/journal/ijerph/instructions#manuscript)

Comment: Despite the fact that the methodology, data analysis, results and introduction are well defined this study lacks an analytical definition of discussion. There is not an equal definition among all the parts of the study.

Response: Thank you for the comment. Unfortunately, the suggestion you make in this comment is not clear to the authors. As noted above, we have chosen to combine the results and discussion sections of this manuscript based on guidelines of the journal. We feel that this was the most appropriate method to contextualize the findings of this study.

Reviewer 3 Report

a) Numbering of figures is not consistent :

       - between lines 115-116 - figure 1 and between lines 259-260 - figure 1 again;

        - 165-166 - figure 2 and 302-302 - figure 2 again;

       - 238-239 - figure 3 and 312-313 - figure 3 again;

       - etc; all figures after line 242 must be renumbered accordingly;

b) Citation of / reference to figures in the main text is not consistent and adequate, since the figures have not been numbered consistently. Please review the entire text and adequately refer to the corresponding figures.

c) Line 281 - "Figure 33-5" ? please reformulate;

d) The methodology is rather limited. It would have been useful (if possible) to have some actual data regarding lead contamination at least of some of the buildings (measurement of lead levels in paint applied throughout the buildings taken into analysis), in order to validate the methology. Please comment in detail on this aspect in subsection 3.5. Limitations.

Author Response

Thank you for the opportunity to submit revisions to our manuscript. We appreciate the time and effort that the editors and reviewers have dedicated to providing meaningful feedback on the manuscript. We have incorporated changes throughout the manuscript to reflect the comments/suggestions provided by the reviewers. Below, please see our responses to your comments and concerns.

Comment:

a) Numbering of figures is not consistent :

       - between lines 115-116 - figure 1 and between lines 259-260 - figure 1 again;

        - 165-166 - figure 2 and 302-302 - figure 2 again;

       - 238-239 - figure 3 and 312-313 - figure 3 again;

       - etc; all figures after line 242 must be renumbered accordingly;

b) Citation of / reference to figures in the main text is not consistent and adequate, since the figures have not been numbered consistently. Please review the entire text and adequately refer to the corresponding figures.

c) Line 281 - "Figure 33-5" ? please reformulate;

Response to comments a) through c): We have rechecked the figure numbers and references to figures throughout the manuscript.

Comment: The methodology is rather limited. It would have been useful (if possible) to have some actual data regarding lead contamination at least of some of the buildings (measurement of lead levels in paint applied throughout the buildings taken into analysis), in order to validate the methodology. Please comment in detail on this aspect in subsection 3.5. Limitations.

Response: Thank you for your comment. Unfortunately, we are unable to obtain actual lead contamination data for the building. We have discussed this limitation in Section 3.5. Limitations.

Lines 424-433: “Another limitation of this study is that due to lack of data availability, the methodology could not be validated using data for lead test in buildings or blood lead level test of children in the County. However, a previous study conducted in Toledo, Ohio confirmed the link between age and value of the residential properties and lead test results for children ages 0-72 months between 2010–2014. The area with a higher concentration of older and lower-valued properties was found to have a significantly higher positive elevated blood lead level test rate [48]. Therefore, the methodology presented in this study is expected to approximate communities of interest where testing efforts should be better targeted.”

Reviewer 4 Report

Review of  ijerph-1115329 Measuring lead paint hazard a

The title is too long and grammatically incorrect.

Line 14. The housing profession is not listed as an audience for this paper.

The abstract does not seem to contain any actual results, only background, methods and recommendations.

Line 37. Title X of the 1992 Housing and Community Development Act is the nation’s most significant lead paint poisoning prevention law, but it is not mentioned at all in this article. The 1971 Act largely failed because it emphasized blood lead screening, not abatement, and placed authority in health departments instead of housing. HUD’s lead paint hazard control program is also not mentioned at all.

Line 61. This sentence seems to say all houses are affected, but how can it be both universal and exist in “many” homes: “universal environmental challenge in many homes…”

Line 65. Reference 17 is outdated and should be replaced by this: https://www.hud.gov/sites/documents/AHHS_REPORT.PDF

Line 128. Actually, housing unit level is the smallest area not census tract: “Census block, the smallest of the three, provides the finest geographic resolution and detail.” While this might not be available, because housing improvements are done on a unit-by-unit basis, census tract information is merely an estimate.

Is there a reason why income and blood lead levels were not included in the mapping?

Line 290. Again, housing is not included in the mix of professions needed.

Many communities have developed such maps. It would be helpful for the authors to state what is new in their analysis.

The English needs to be improved throughout this paper.

Author Response

Thank you for the opportunity to submit revisions to our manuscript. We appreciate the time and effort that the editors and reviewers have dedicated to providing meaningful feedback on the manuscript. We have incorporated changes throughout the manuscript to reflect the comments/suggestions provided by the reviewers.

Below, please see our responses to your comments and concerns.

Comment: The title is too long and grammatically incorrect.

Response: Thank you. We have edited the title to “Neighborhood-level Lead Paint Hazard for Children Under 6: A Tool for Proactive and Equitable Intervention.”

Comment: Line 14. The housing profession is not listed as an audience for this paper.

Response: We have mentioned “planners” (Line 13) and “local housing and planning authorities” (Lines 23-24).

Comment: The abstract does not seem to contain any actual results, only background, methods and recommendations.

Response: We added highlights of results:

Lines 16 – 20: “Results highlight areas of potential lead paint hazard hotspots within the county in a midwestern state studied, which coincides with higher concentration of non-white children. This places lead paint hazard in the context of social determinants of health, where existing disparity in distribution of social and economic resources reinforces health inequity.”

Comment: Line 37. Title X of the 1992 Housing and Community Development Act is the nation’s most significant lead paint poisoning prevention law, but it is not mentioned at all in this article. The 1971 Act largely failed because it emphasized blood lead screening, not abatement, and placed authority in health departments instead of housing. HUD’s lead paint hazard control program is also not mentioned at all.

Response: Thank you for the suggestion. We have edited the manuscript to include Title X of the 1992 Housing and Community Development Act.

Lines 38-43: “In 1992, Congress passed the Residential Lead-Based Paint Hazard Reduction Act of 1992, also known as Title X of the Housing and Community Development Act. This law directed US Department of Housing and Urban Development and US Environmental Protection Agency to require the disclose of known information on lead-based paint and lead-based paint hazards before the sale of lease of most housing built before 1978 [12].”

Comment: Line 61. This sentence seems to say all houses are affected, but how can it be both universal and exist in “many” homes: “universal environmental challenge in many homes…”

Response: Thank you. The sentence was edited.

Lines 63-64: “However, lead-based paint hazards remain to be a critical environmental hazard in many homes of children across the United States”

Comment: Line 65. Reference 17 is outdated and should be replaced by this: https://www.hud.gov/sites/documents/AHHS_REPORT.PDF

Response: Thank you for the information. We have replaced the reference with the one suggested.

Comment: Line 128. Actually, housing unit level is the smallest area not census tract: “Census block, the smallest of the three, provides the finest geographic resolution and detail.” While this might not be available, because housing improvements are done on a unit-by-unit basis, census tract information is merely an estimate.

Response: We have in fact used housing unit level data (parcel level data) to estimate lead hazard risk based on age, value, and quality of residential property (section 2.2.1). Three census geographic units (block, block group, and tract) were then used for aggregating parcel level lead hazard risk at different geographic levels.

Comment: Is there a reason why income and blood lead levels were not included in the mapping?

Response: Unfortunately, blood lead level data and income at the household level were not available. While income certainly plays a role in the type of housing families (and ultimately children) have access to, our methodology for determining neighborhood lead risk is based on factors that directly related to the home (age, value and building grade) that are known to be correlated with lead risk. However, in our analysis comparing lead hazard to social determinants of health and opportunity indices (see Section 3.3), we compare our lead risk maps to other indices like the Child Opportunity Index (COI) and the Health Opportunity Index (HOI). Both the COI and HOI include economic indicators in their indices.

Comment: Line 290. Again, housing is not included in the mix of professions needed.

Response: We have added ‘local housing authorities’ and also discussed how ‘city planners’ can benefit from lead risk maps for focused housing renovation in the same paragraph.

Lines 291-296: “In the process, the placed-based approach suggested here can also support cross-sectional collaboration in the identification of multiple parties to be involved – parents, caregivers, healthcare providers and professionals, community organizers, and local housing authorities – and development of partnerships among them, with a shared goal of improving the health of children in the neighborhoods [23,34-37].”

Comment: Many communities have developed such maps. It would be helpful for the authors to state what is new in their analysis.

Response: The method presented in the manuscript used housing or parcel level data in estimating lead hazard risk based on multiple indicators (i.e. age, value, and quality of the parcel). Risk scores were then aggregated at different census geographic levels for visualization and this enabled analysis of racial/ethnic inequity in lead hazard at neighborhood level. We believe that this is the first study to the authors’ knowledge that used this method for estimating neighborhood level lead risk.

Comment: The English needs to be improved throughout this paper.

Response: The manuscript was reviewed and edited by native English-speaking authors.

Round 2

Reviewer 3 Report

The suggestions regarding the content and form of the manuscript were taken into consideration and the manuscript has been revised.